# Acoustic Stimulation of Anaerobic Digestion: Effects on Biogas Production and Wastewater Malodors

**John Loughrin** *, **Philip Silva** , **Nanh Lovanh and Karamat Sistani**

Food Animal Environmental Systems Research Unit, Agricultural Research Service, United States Department of Agriculture, 2413 Nashville Road, Suite B5, Bowling Green, KY 42101, USA
* Correspondence: john.loughrin@usda.gov; Tel.: +1-270-781-2260

**Abstract:** Wastewater created from various solid wastes and agricultural residues was treated by anaerobic digestion, and the biogas and wastewater odors were quantified. One digester was exposed to low-frequency sound (<5 kHz) from underwater loudspeakers, while the other received no sonic treatment. It was hypothesized that low-frequency sound, by accelerating the breakdown of sludge via mechanisms such as cavitation induction and mechanical vibration, and enhancing biogas production, could also affect the concentrations of wastewater odors. During warm seasons, biogas production from the sound-treated digester was 29% higher than that from the control digester, and 184% higher during the cool season. Malodors—Mainly consisting of typical aromatic malodorants such as *p*-cresol and skatole, aliphatic secondary ketones, and dimethyl disulfide—were quantified. In contrast to the findings for biogas production, little difference was found in the concentrations of volatile compounds in the control and sound-treated digestates. Concentrations of dimethyl polysulfides increased over time in both the control and sound-treated digestates, likely due to the use of recycled system effluent that contained precipitated elemental sulfur. The digestate contained considerable concentrations of volatile fatty acids and ammonium, but due to the near neutral pH of the digestate it was surmised that neither made appreciable contributions to the wastewater's malodor. However, the volatile fatty acid concentrations were reduced by sonic treatment, which was not unexpected, since volatile fatty acids are precursors to methane. Therefore, although sonic treatment of the anaerobic digestate boosted biogas production, it did not markedly affect the wastewater malodors. The biosynthetic origins of wastewater malodors are discussed in this paper.

**Keywords:** anaerobic digestion; biogas; carbon dioxide; greenhouse gas; malodor; methane; sulfide; sonication; ultrasonication

## 1. Introduction

Anaerobic digestion of wastewater and sludge is often touted for its potential for energy production and its ability to reduce greenhouse gas emissions [1]. In addition, by reducing the quantity of organic matter in waste, pollution of ground and surface waters is reduced, and nutrients such as nitrogen and phosphorus are more easily recovered to further aid in reducing eutrophication. By operating digesters at thermophilic (>50 °C) temperatures, reductions in pathogens may also be accomplished [2].

Furthermore, anaerobic digestion may be an effective means of reducing waste malodors. However, to achieve significant reductions in malodors, it is often necessary to operate digesters at elevated temperatures. For instance, Orzi et al. [3] studied 10 full-scale digesters, which achieved a 98.8% overall reduction in digestate malodor, as determined by panelists in an olfactometer analysis. Variation in hydraulic retention times (HRTs) from 20 to 70 d did not have a pronounced influence on malodor reduction, but all of the digesters were operated at 39 °C or above. Welsh et al. [4] found that anaerobic digestion at 35 °C was more effective than 25 °C in reducing the offensiveness of swine manure. Wilson et al. showed that even in thermophilic digestion (>50 °C), slight increases in temperature

reduced the concentrations of volatile sulfur compounds such as methanethiol, dimethyl sulfide, dimethyl disulfide, and dimethyl trisulfide [5]. Consequently, elevated temperature seems to be more effective in reducing wastewater malodors than long HRT. The city of Los Angeles, CA, found that operating digesters at thermophilic ranges reduced biosolids' odor content by 70% as compared to biosolids produced at mesophilic temperatures, but it was unclear whether some of the improvement was due to volatilization of the malodors, since the city of Tacoma, WA, found that potentially dangerous gas-phase concentrations of $H_2S$ could be reduced at wastewater treatment plants by lowering the digester temperatures from 46 to 38 °C [6].

Recently, we demonstrated that the treatment of anaerobic digesters with sound at sonic frequencies (<20 kHz) could be used to accelerate biogas production and sludge degradation [7,8]. Sound was provided to 9.1 $m^3$ of anaerobic digestate by means of waterproofed loudspeakers on a continuous 2 h on/1 h off cycle. In a first trial, the biogas production was more than double that of a control digester, while in a second trial the biogas production was increased by 27% over that of the control during warm weather, and by 74 times that of the control in cold weather. Given that the sonic treatment seemed to some extent to substitute for wastewater heating in accelerating biogas production, this study examines whether sonic treatment of anaerobic digestate can also have a positive effect on wastewater malodors. Digestate samples collected from the primary anaerobic digesters in the previous studies were analyzed for volatile organic compounds (VOCs). Samples were collected during periods of warm weather—from May 2018 to October 2018, from June to July 2020, and from October 2019 to March 2020.

## 2. Materials and Methods

### 2.1. Digester Description

The feeding schedule of the digesters has been described previously [7,8]. In 2018, the digesters were mostly fed cracked corn, with occasional supplementation with defatted soybean meal, after initial seeding of the digesters with swine waste, chicken litter, and digestate from a commercial anaerobic digester operated at a range of 46–50 °C. Meanwhile, in 2019–2020, the digesters were fed a varying mixture of cracked corn, waste activated sludge from the Bowling Green KY municipal wastewater treatment facility, corn stover, and wastepaper/cardboard. The digesters were fed either once or twice weekly using a volume of 800–900 L. The digesters therefore had an HRT of approximately 12 weeks when fed once weekly, and about 6 weeks when fed twice weekly. The wastewater outlet from the digesters was approximately one meter above the bottom and, therefore, given the settling of solids, was designed to uncouple the sludge retention time from the wastewater retention time.

Figure 1 shows the amounts of volatile solids (VS) fed per month, along with the monthly average daily temperature and monthly average solar insolation. The digesters were fed once or twice weekly with the feed mixed with 800 L of recycled effluent from the system. The digesters were not fed from November 2018 through April 2019. Data for temperature and insolation were obtained from the Warren County Kentucky Mesonet site located on the Western Kentucky University Farm (kymesonet.org, accessed on 11 August 2022).

In 2018, the digestate was exposed to sound from waterproofed Skar Audio FSX8 8-inch (20.3 cm) 4 Ω speakers rated at 175 W RMS (root-mean-square) power (Skar Audio, Tampa, FL, USA), powered by Pyle Audio PTAU 55 stereo amplifiers rated at 120 W RMS per channel (Pyle Audio, Brooklyn, NY, USA). In 2019–2020, the speakers were replaced with Skar Audio FSX10-4 10-inch (25.4 cm) 4 W speakers rated at 200 W RMS. Details of the sound files and recording equipment used were published previously [4,5]. The stereo was operated on a 2 h on/1 h off continuous cycle.

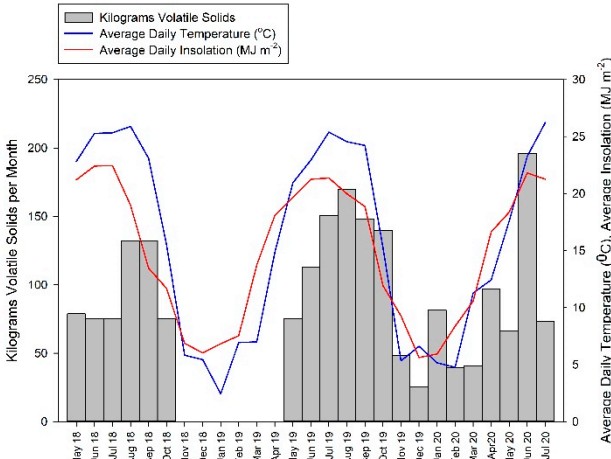

**Figure 1.** Amounts of volatile solids fed per month to each of the digesters, with average daily temperature and average daily insolation received for each month of the study.

### 2.2. Volatile Compound Extraction and Quantification

Table 1 lists the volatile compounds identified and quantified in the anaerobic digestate of the control and sound-treated digesters, along with selected physical properties. A number of other compounds—such as dimethyl trisulfide, dimethyl tetrasulfide, and dimethyl pentasulfide—were not quantified, due to a lack of authentic standards for quantification. In addition, other compounds that are important components of wastewater malodor— such as hydrogen sulfide, methyl mercaptan, and dimethyl sulfide—were not detected, due to their being too volatile for analysis by stir-bar sorptive extraction (SBSE) (with boiling points of −60.2, 5.95, and 27 °C, respectively) and/or too polar (with log octanol partition coefficients of 0.78 and 0.977 for methyl mercaptan and dimethyl sulfide, respectively).

**Table 1.** Selected physical properties of compounds quantified in this study (listed in elution order on a DB-5 gas chromatography column [1]).

| Compound | Molecular Weight (g mol$^{-1}$) | Boiling Point (°C) | Log P |
|---|---|---|---|
| Dimethyl disulfide | 94.19 | 109.8 | 1.77 |
| Toluene | 92.14 | 110.6 | 2.73 |
| 2-Hexanone | 100.16 | 127.6 | 1.38 |
| 2-Heptanone | 114.18 | 151.0 | 1.98 |
| Benzaldehyde | 106.12 | 178.1 | 1.64 |
| Oct-1-en-3-ol | 128.21 | 175.0 | 2.60 |
| Phenol | 94.11 | 181.8 | 1.46 |
| *Para*-Cresol | 108.14 | 201.9 | 1.94 |
| 2-Nonanone | 142.24 | 195.3 | 3.14 |
| Benzyl acetate | 150.18 | 213.0 | 1.96 |
| *Para*-Ethylphenol | 122.17 | 217.9 | 2.58 |
| Indole | 117.15 | 254.0 | 2.14 |
| Skatole | 131.18 | 266.0 | 2.60 |
| 2-Tridecanone | 198.35 | 263.0 | 4.68 |

[1] Data obtained from The National Institutes of Health, chemidplus Advanced [9].

Digestate samples were taken from the digesters approximately 1 m above the bottom of the tank by means of a three-way valve, and stored at −20 °C until analyzed. Volatile compounds in the digestate were quantified by extraction via stir-bar sorptive extraction (SBSE) using Twister stir bars (10 mm by 0.32 mm) with a 1 mm thick polydimethylsiloxane (PDMS) coating (Gerstel, Baltimore, MD, USA), which had been preconditioned at 250 °C for 40 min under a stream of high = purity He. The stir bars were placed in sealed vessels containing 100 mL of digestate and extracted for 2 h while stirring at 150 rpm. After

extraction, the stir bars were removed, rinsed with deionized water, and blotted dry. They were then placed in 8.9 cm long by 6.35 mm o.d. thermal desorption tubes (Markes International, Sacramento, CA, USA) and desorbed as described below.

Gas chromatography-mass spectrometry (GC-MS) of semi-volatile odor compounds was performed using a Shimadzu Nexis GC-2030/QP2020 NX single-quadrupole mass spectrometer interfaced with a TD-30 thermal desorption system (Shimadzu Scientific Corp., Columbia, MD, USA). The stir bars were placed in the glass desorption tubes and desorbed at 280 °C for 5 min with a He flow rate of 30 mL min$^{-1}$. The cryogenic trap was maintained at $-20$ °C, with a transfer line temperature of 250 °C. The GC was equipped with a 30 m by 0.25 mm Rxi-5 ms (95% PDMS, 5% diphenyl polysiloxane) column with a film thickness of 0.25 μm (Restek Corp., Bellefonte, PA, USA). The GC operating conditions were an initial oven temperature of 50 °C for 1 min, then increased at 1.5 °C min$^{-1}$ to 105 °C, and thereafter increased at 15 °C min$^{-1}$ to 210 °C, with a final hold time of 2 min, and using a linear He flow velocity of 47.2 cm s$^{-1}$. The mass spectrometer used a scanning range of 35–500 m/z and a scan speed of 1666 amu s$^{-1}$. Identification and external calibration of compounds was based on comparison of spectra and retention times with those of authentic standards (Sigma-Aldrich Inc., St. Louis, MO, USA).

Biogas production was measured using diaphragm/bellows positive displacement meters (EKM Metering, Santa Cruz, CA, USA), and biogas quality was measured by gas chromatography (GC), as previously described [7,8]. Gas volumes were reported at ambient conditions of temperature and pressure, and $CO_2$ and $CH_4$ concentrations were quantified from samples taken by withdrawing 25 mL samples from an outlet upstream of the gas meter by means of a three-way valve and injecting the sample into a 20 mL vial equipped with a rubber septum.

Methane and $CO_2$ were analyzed on a Varian Model CP-3800 (Agilent Technologies, Santa Clara, CA, USA) GC modified for greenhouse gas (GHG) analysis. The GC was equipped with a model 1041 on-column injector operated at 75 °C and 263 kPa, which was connected to a 10-port gas-sampling valve and pressure-actuated solenoid valve. Half a milliliter of the vial headspace was injected using a syringe temperature of 35 °C and needle residence time of 30 s, with 250 μL of the sample transferred onto a 1.8 m by 1.6 cm o.d. column packed with 80/100 mesh Hay Sep Q (Agilent Technologies), with a He flow rate of 55 mL min$^{-1}$. The column was connected to a thermal conductivity detector (TCD) operated at 120 °C, and with a filament temperature of 200 °C, for $CO_2$ analysis. From the TCD, the sample went to a flame ionization detector operated under the following conditions: $N_2$ makeup gas 15 mL min$^{-1}$, $H_2$ 30 mL min$^{-1}$, air 300 mL min$^{-1}$, and a detector temperature of 275 °C. The GC column oven was operated at an initial temperature of 50 °C for 4 min, and then increased at 50 °C min$^{-1}$ to 100 °C and held at this temperature for 1 min. Four point calibrations were performed for $CO_2$ and $CH_4$.

## 3. Results

### 3.1. Biogas Production

During the first trial of the digesters in the warmer months of 2018, the digestate temperature in the control digester and sound-treated digester averaged $27.9 \pm 5.3$ and $26.9 \pm 4.7$ °C, respectively. In the second trial, from late spring 2019 to summer 2020, the average digestate temperature of the control and sound-treated digesters was $20.2 \pm 9.4$ and $20.4 \pm 8.6$ °C, respectively, with maximum temperatures of 33.6 and 31.8 °C, respectively, and minimum temperatures of 3.5 and 3.9 °C, respectively. Since the temperature range for mesophilic digestion may be considered to range from 35 to 40 °C [10], the digesters were operated at quite low temperatures.

In the year 2018, gas production was measured from mid-April to early October (Figure 2). Average weekly gas production over this period was 87% higher from the sound-treated digester (11.9 m$^3$ wk$^{-1}$) than the control digester (6.4 m$^3$ wk$^{-1}$), and the concentrations of $CO_2$ and $CH_4$ averaged 10,700 and 36,800 mmol m$^{-3}$, respectively, for the control digester, and 11,100 and 38,200 mmol m$^{-3}$, respectively, for the sound-treated

digester, from early July to mid-October [7]. Although the gas pressures in the control and sound-treated digesters were not continually monitored, the fact that the sound-treated digester produced more gas than the control digester necessarily means that the back pressure in the sound-treated digester was higher, and resulted in faster filling and emptying of the gas meter bellows. Spot checks on the digesters indicated back pressures of approximately 9–12 mm Hg (1.2–1.6 kPa). Higher back pressure on the sound-treated digester would account for higher $CO_2$ and $CH_4$ gas concentrations in the sound-treated digester as compared to the control digester. The higher gas production rate was likely a consequence of higher microbial activity in that tank due to sonic disruption of the sludge facilitating microbial colonization, as discussed below.

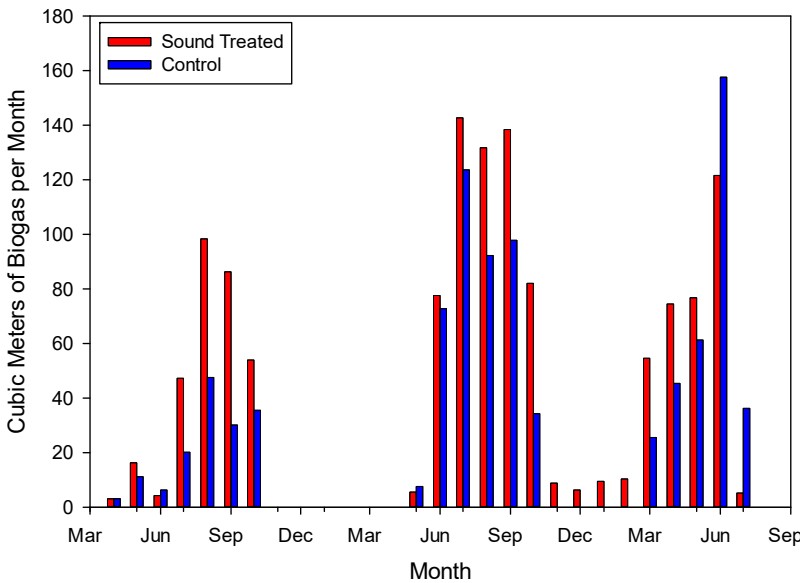

**Figure 2.** Average monthly gas production from the sound-treated and control anaerobic digesters.

In the second trial, lasting from late May 2019 to mid-July 2020, the average weekly gas production was 12.9 $m^3$ $wk^{-1}$ and 15.7 $m^3$ $wk^{-1}$ for the control and sound-treated digesters, respectively [8]. The carbon dioxide concentrations averaged 10,300 mmol $m^{-3}$ and 10,900 mmol $m^{-3}$ for the control and sound-treated digesters, respectively, whereas the $CH_4$ concentrations averaged 27,400 mmol $m^{-3}$ and 38,900 mmol $m^{-3}$ for the control and sound-treated digester, respectively. The $CH_4$ concentration percentage averaged 77.5% in 2018 and 72.7% in 2019–2020. The slightly lower average gas quality in the later evaluation period was likely due to its encompassing both warm and cool seasons, and to a more varied feedstock, which included wastepaper and waste activated sludge. From May 2018 through June 2020, monthly $CH_4$ production averaged 209 g $kg^{-1}$ VS for the control digester and 326 g $kg^{-1}$ VS for the sound-treated digester.

From June 2018 through July 2020, the mean specific biogas yields from the control and sound-treated digesters were 0.39 ± 0.31 and 0.60 ± 0.39 $m^3$ $kg^{-1}$ VS, respectively. Towards the end of the evaluation period in the summer of 2020, the control digester outperformed the sound-treated digester, producing an apparent yield of 0.8 and 0.49 $m^3$ $kg^{-1}$ VS in June and July, respectively whereas the sound-treated digester produced an apparent yield of only 0.62 and 0.07 $m^3$ $kg^{-1}$ VS, respectively, during the same period. Given that the retention of solids was decoupled from HRT, this was doubtless due to the previous depletion of VS in the sound-treated digester. From October 2019 through April 2020, the apparent biogas yield was 0.19 ± 0.26 and 0.50 ± 0.44 $m^3$ $kg^{-1}$ VS for the control and sound-treated digesters, respectively. We specifically use the phrase apparent biogas yield in that the biogas yield during any given period is not only reflective of the current feeding rate, but is also dependent on the digestion of previously accumulated feed. The sonic treatment during the cooler months enhanced gas production significantly relative to the

control digester. This necessarily meant that more VS were depleted in the sound-treated digester than in the control digester. This likely explains the lower biogas production by the sound-treated digester in the last two months of the experiment. Thus, by uncoupling the retention of solids from HRT, and retaining solids in the digesters, the sonic treatment was able to deplete VS compared to the control digester, accounting for its low gas production in the last two months of the experiment. In May 2020, it was found that sludge from the sound-treated digester contained 46.3 g VS $kg^{-1}$ sludge, while sludge from the control digester contained 95.2 g VS $kg^{-1}$ sludge [8].

### 3.2. Digestate Malodors

Fourteen odorous organic compounds were quantified from the digestate of the sound-treated and control anaerobic digesters (Table 2). Most of these compounds were selected due to their high concentrations and their status as important components of wastewater malodor [11,12]. Other compounds quantified but not occurring in high concentrations were not typical wastewater malodors. These included benzaldehyde—Which is a component of many fruit aromas such as cherry, peach, and guava [13–15]—and benzyl acetate, which is a common floral fragrance compound [16,17]. It is likely that benzaldehyde was formed by reduction of benzoic acid—A common natural product of plants. The reduction/hydrogenation of organic matter in anaerobic environments is common, and can be due to the organic matter serving as a terminal electron acceptor for anaerobic respiration [18]. A further reduction to benzyl alcohol would result in the alcohol moiety of benzyl acetate.

**Table 2.** Concentrations of malodorous compounds identified in anaerobic digestate [1].

| | Treatment | |
|---|---|---|
| **Compound** | **Control** | **Sound-Treated** |
| | Concentration (Milligrams per Liter) | |
| Dimethyl disulfide | 171 ± 33.1 | 173 ± 29.6 |
| Toluene | 4.61 ± 2.1 | 6.0 ± 1.93 |
| Phenol | 97.5 ± 17 | 70.4 ± 12.4 |
| *para*-Cresol | 460 ± 116 | 435 ± 128 |
| *para*-Ethylphenol | 49.4 ± 25.3 | 50.9 ± 25.4 |
| Indole | 141 ± 30.2 | 104 ± 22.8 |
| Skatole | 116 ± 51.6 | 111 ± 17.2 |
| | Concentration (Micrograms per Liter) | |
| 2-Hexanone | 951 ± 349 | 581 ± 254 |
| 2-Heptanone | 835 ± 240 | 567 ± 161 |
| 2-Nonanone | 320 ± 119 | 347 ± 91 |
| 2-Tridecanone | 469 ± 333 | 717 ± 297 |
| 1-Octen-3-ol | 620 ± 379 | 452 ± 298 |
| Benzaldehyde | 41 ± 25 | 29 ± 14 |
| Benzyl acetate | 19 ± 12 | 32 ± 14 |
| Total identified | 1043 | 953 |

[1] Data represent the mean ± standard error of the mean of 14 averaged monthly determinations encompassing May to October 2018, October 2019 to March 2020, and June to July 2020.

Oct-1-en-3-ol is a common fungal odorant, and the (*R*)- enantiomer is a characteristic impact compound responsible for the odor of mushrooms [19]. Oct-1-en-3-ol is generated via peroxidation of linoleic acid by lipoxygenase, with the hydroperoxide cleaved by hydroperoxide lyase to form 1-octen-3-ol. Lipoxygenases are common in eukaryotes, including fungi, but rare in bacteria [20].

### 3.3. Secondary Ketones

The secondary ketones are likely synthesized by β-oxidation of free fatty acids. In β-oxidation, a fatty acid is degraded to acetyl-CoA thioesters, as well as generating the reduced coenzymes $FADH_2$ and NADH. β-Oxidation of unsaturated fatty acids requires additional enzymatic steps that are not shown in this scheme [21].

$$\text{(scheme)} \tag{1}$$

After a variable number of flavoprotein-mediated redox cycles producing acetyl-CoA and a fatty acyl-CoA thioester (1), the cycle can be terminated by a thioesterase to produce free coenzyme A (CoASH) and a β-keto acid (2). The β-keto acid is unstable, and likely spontaneously loses $CO_2$ to form a methyl ketone [22].

$$\text{(scheme)} \tag{2}$$

The β-oxidation of odd-numbered fatty acids could account for the presence of the odd-numbered secondary ketones [23]. Although the breakdown of fatty acids to acetyl-CoA and a fatty acyl-CoA thioester is quite efficient, the alternative pathway that results in a free β-keto acid still happens frequently, and has been enhanced in *Escherichia coli* by chromosomal deletion of the enzyme responsible for cleavage of the β-keto thioester, along with overexpression of the thioesterase responsible for the formation of the β-keto acid [22]. The engineered bacteria were capable of causing a 700-fold increase in methyl ketone production.

The secondary ketones were present at relatively low concentrations—less than 1 mg $L^{-1}$ of digestate. Sonic treatment of the anaerobic digestate did not markedly affect the concentrations of secondary ketones, given that the concentrations of these compounds were similar in both digesters (Table 2). 2-Heptanone and 2-nonanone have low odor detection thresholds [24] (4.8 and 5.5 ppb in air, respectively), and so it is likely that these compounds had a contribution to the odor of the wastewater. During warm weather, the summed concentration of the four secondary ketones averaged 3.2 µg $L^{-1}$ digestate in the control digester, and 2.9 µg $L^{-1}$ in the sound-treated digestate. During cool weather, the four secondary ketones averaged 2.8 and 1.7 µg $L^{-1}$ in the control and sound-treated digestate, respectively. The higher concentrations of secondary ketones during warm seasons were likely due to greater microbial growth during these periods.

### 3.4. Aromatic Malodorants

Toluene and *p*-cresol are synthesized from L-phenylalanine and L-tyrosine, respectively, as shown in Equation (3). Following transamination of an acceptor, α-keto acid forms 2-oxo-3-phenylpropanoic acid or 2-oxo-3-(4-hydroxy) phenylpropanoic acid. Following decarboxylation of the 2-oxoacid, phenyl acetate (or 4-hydroxyphenylacetate) is either decarboxylated, or first reduced to the aldehyde and then decarbonylated to form the final product [25,26]:

$$\text{(scheme)} \tag{3}$$

Skatole is produced similarly from indole acetate:

(4)

It is interesting to note that the enzymes that produce toluene, *p*-cresol, and skatole belong to the family of glycyl-radical-activating enzymes produced by facultative and strict anaerobes such as the Clostridiales, Actinomyceota, Enterobacteriaceae, and Fusobacteriaceae [25]. These enzymes are easily inactivated by oxygen [27].

Phenol is formed by bacteria such as *Clostridium tetanomorphum* via the action of L-tyrosine ammonia lyase to produce pyruvic acid and ammonium [28,29]:

(5)

Similarly, indole is produced from tryptophan by L-tryptophanase [30]:

(6)

4-Ethylphenol has been shown to be produced by the decarboxylation of p-coumaric acid by certain yeasts and lactic acid bacteria to produce 4-vinylphenol, which is then reduced to 4-ethylphenol [31,32]:

(7)

Given the persistence of these aromatic malodorants in stored animal wastes, they have been referred to as dead-end products of amino acid catabolism—or at least as produced by intestinal bacteria [33] These aromatic malodorants have been shown to be degraded by sulfate-reducing bacteria [34]—aerobically by the purple non-sulfur bacterium *Rhodopseudomonas palustris* [35], and anaerobically by *Clostridia* spp. and by methanogenic consortia [36,37]. The intermediates in the mineralization of indole and skatole are isatin (1-H-indole-2,3-dione) and 3-methyloxindole (3-methyl-1,3-dihydroindole-2-one), respectively [37].

In the early stages of digester startup, the concentrations of aromatic malodorants were high, and in May 2018 *p*-cresol averaged 639 and 960 mg $L^{-1}$ in the control and sound-treated digesters, respectively. By September, the concentrations of *p*-cresol had declined to 108 and 220 mg $L^{-1}$ in the control and sound-treated digesters, respectively. Thereafter, the *p*-cresol concentrations remained low, and even during the winter of 2019–2020, when biogas production dropped, the *p*-cresol concentrations never exceeded 470 and 307 mg $L^{-1}$ in the control and sound-treated digesters, respectively. Other aromatic malodorants behaved similarly. Since the concentrations of aromatic malodorants declined considerably after digester startup, it is likely that the aromatic malodorants were catabolized by the methanogenic consortia of the digesters.

The aromatic malodorants are major contributors to wastewater malodor. Still, the odor detection thresholds vary widely for these malodorants, ranging from 330 ppb (parts per billion) in air for toluene to 54 and 5.6 ppt (parts per trillion) in air for *p*-cresol and

skatole, respectively [38]. During warm weather, the aromatic malodors averaged 1350 and 1260 mg L$^{-1}$ in the control and sound-treated digestates, respectively, and during cool weather they averaged 582 and 402 mg L$^{-1}$ in the control and sound-treated digestates, respectively.

*3.5. Volatile Sulfur Compounds*

Dimethyl sulfide (DMS) is formed from L-methionine via dimethylsulfoniopropionate (DMSP). Although formerly thought to be exclusively produced via a largely aerobic and light-dependent process by oceanic dinoflagellates, a large number of bacteria are now known to produce DMSP and DMS, as well as methanethiol [39]. Methanethiol is a weak acid that forms methylthiolate, which is a strong nucleophile. It reacts with elemental sulfur (S$_8$) to form methyl polysulfide [40]:

$$CH_3S^- + S_8 \leftrightarrow CH_3S_9^- \tag{8}$$

Methyl polysulfides are unstable. They decompose to shorter dimethyl polysulfides and polysulfides, with $x + y = 10$:

$$CH_3S^- + CH_3S_9^- \leftrightarrow CH_3S_xCH_3 + S_y^{2-} \tag{9}$$

Jocelyn et al. found that H$_2$S and dimethyl disulfide (DMDS) are formed if S$_8$ is present in limited quantities, and that longer-chain sulfides are formed if S$_8$ is not limited [41]. Ver Leerdam et al. found that the reaction of methanethiol with elemental sulfur occurs under aeration [42]. In our system, wastewater from a final-stage aeration tank was mixed with feed and fed back to the digesters. Oxidation of H$_2$S by photosynthetic bacteria can produce octacyclic elemental sulfur [42], and wastewater gas chromatographic/mass spectral analysis of the terminal aeration tanks registered an average of $9.0 \times 10^6$ and $13 \times 10^6$ S$_8$ ion counts for the control and sound-treated systems, respectively, while S$_8$ was undetectable in the primary digesters. Although elemental sulfur did not chromatograph well on the GC column we used, this finding serves to confirm that elemental sulfur was formed in the aeration tanks.

Due to the limitations of our methodology, we were unable to quantify methanethiol and dimethyl sulfide, and did not have standards of the higher homologues of DMDS, dimethyl trisulfide (DMTS), dimethyl tetrasulfide (DMTetS), or any of the longer-chain-length sulfides. The digestates contained considerable concentrations of DMDS, and its concentration was virtually identical in the control and sound-treated digestates (Table 2). Using mass spectral total ion counts to track changes in the concentrations of the volatile disulfides (Figure 3), we found that their concentrations increased markedly in 2019–2020 as compared to their 2018 concentrations.

Since the aeration tank at the end of the system contained S$_8$, but the primary anaerobic digester did not, it is likely that recycling the elemental sulfur back to the system enhanced the production of dimethyl polysulfides. This finding shows that it would be desirable to remove elemental sulfur from the wastewater before recycling it back to the system, in order to reduce the concentrations of dimethyl polysulfides and H$_2$S.

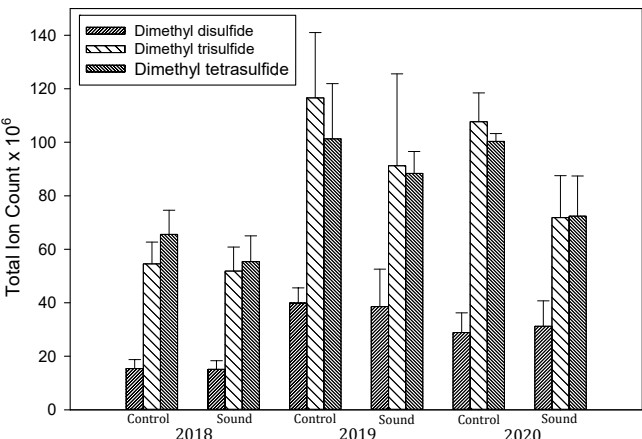

**Figure 3.** Averaged total ion counts for dimethyl sulfide, dimethyl trisulfide, and dimethyl tetrasulfide in anaerobic digestate, 2018–2020. Data represent the mean $\pm$ standard error of the mean.

*3.6. Volatile Fatty Acids and Ammonium*

Volatile fatty acids (VFAs) are largely formed via the fermentation of carbohydrates by prokaryotes to produce ATP in the absence of oxygen. Acetic acid may also be produced by acetogens via the reduction of two molecules of $CO_2$ with four molecules of $H_2$ [43]. The longer-chain VFAs are converted to acetate, and are then consumed by methanogens via cleavage of acetate to $CH_4$ and $CO_2$, whereas other methanogens produce $CH_4$ via reduction of $CO_2$ with $H_2$ [44].

Total VFA concentrations (acetic, propanoic, butyric, and *iso*-butyric acids) in cool weather (October 2018 through March 2019) were 44% higher in digestate from the control digester than from the sound-treated digester (6600 vs. 4600 $\mu$M), and in warm weather, total VFA concentrations were 89% higher in the control digestate than in the sound-treated digestate (1200 vs. 650 $\mu$M), reflecting consumption of VFAs by methanogens in the sound-treated digester [6]. The highest concentrations of acetate occurred during the winter of 2019–2020, when they averaged 276 mg $L^{-1}$ in the control digester and 208 mg $L^{-1}$ in the sound-treated digester.

During the second trial of the digesters, the pH of the control digester averaged 7.02, while the pH of the sound-treated digester averaged 7.08. However, given an average $pK_a$ of 4.8 for the VFAs [9], over 98% of the VFAs would be in the conjugate base form, which likely limited the contribution of VFAs to the wastewater malodor (Figure 4).

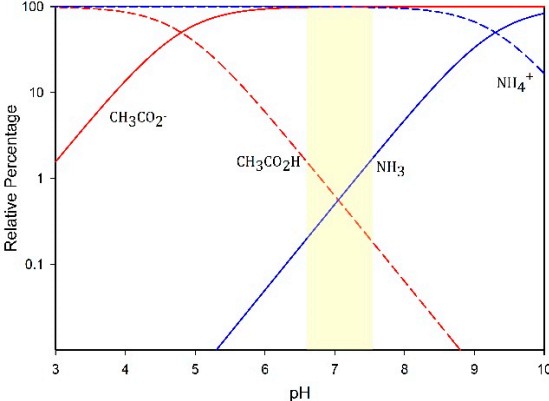

**Figure 4.** Bjerrum plot with the relative concentrations of acetic acid and its conjugate base, and of ammonia and its conjugate acid. The shaded area represents the mean pH of the digesters $\pm$ standard deviation of the mean.

Ammonia concentrations, on the other hand, were virtually identical in the control and sound-treated digestates, averaging 17.8 and 17.9 $\mu$M (321 and 323 mg $L^{-1}$), respectively.

In warm weather, the ammonium concentrations averaged 332 mg $L^{-1}$ in the digestates for both digesters, while in cool weather the ammonium concentrations averaged 255 mg $L^{-1}$. The lower ammonium concentrations in the cool seasons were probably a consequence of lower feeding rates and/or lower microbial activity.

Urease activity is common in the environment. Many microorganisms hydrolyze urea intercellularly into $NH_4^+$ and $HCO_3^-$, and use the resulting proton gradient for ATP synthesis [45]. Thus, it is not surprising that the $NH_4^+$ concentrations in the control and sound-treated digestates were virtually identical. In the near-neutral pH range at which the digesters were operated, $NH_3$ would account for approximately 0.2–1.5% of the acid–base pair. Figure 4 shows a Bjerrum plot diagramming the idealized concentrations of the conjugate acid–base pairs for acetic acid and ammonia.

In any case, the odor thresholds of the VFAs and $NH_3$ are quite high in comparison to other malodorants of anaerobic digestate. Acetic and butyric acids have reported odor thresholds of 6 ppb and 0.19 ppb v/v, respectively, while $NH_3$ has a reported odor threshold of 1.5 ppm v/v. Other typical wastewater malodorants—i.e., DMS, *p*-cresol, and skatole—have reported odor thresholds of 3 ppb, $5.4 \times 10^{-2}$ ppb, and $5.6 \times 10^{-3}$ ppb v/v, respectively [38]. Thus, given the near-neutral pH of the digestate, neither VFAs nor $NH_3$ are likely to make major contributions to malodors compared to other volatile compounds, despite their relatively high concentrations.

### 3.7. Sonic Treatment of Anaerobic Digestate

Sonic treatment of anaerobic digestate works to increase biogas production in the same manner as ultrasonic pretreatment—that is, via sludge disruption. This occurs via various mechanisms, such as acoustic streaming inducing mass flow in the wastewater, vibrational energy imparted to the sludge, and cavitational inception and collapse, which can impart localized pressures of great amplitude [46]. Sludge disruption facilitates microbial colonization of the sludge by solubilizing nutrients and increasing the sludge's surface area [47,48]. This was evidenced by the higher background audio levels in the digester with prior sound exposure, which were likely due to greater gas production by the digester's microflora, resulting in numerous bubbles that act as acoustic resonators [46] (Supplementary Materials). Although ultrasonic pretreatment of sludge and direct sonic treatment share mechanistic similarities, they differ in sound frequency, the timing and duration of exposure, and power requirements.

Pretreatment of sludge with ultrasonic frequencies has been shown to significantly enhance the disintegration of the sludge, and thereby accelerate anaerobic digestion [49–51]. However, attenuation of sound is proportional to the sound's frequency. Largely for this reason, disintegration of waste by ultrasonication is usually conducted at relatively low frequencies—less than 100 kHz, and most often at less than 30 kHz [48]. Furthermore, lower frequencies create larger cavitation bubbles which, when they collapse, produce larger mechanical jet streams, causing greater sludge disintegration [47].

Used as a pretreatment for the disintegration of sludge prior to anaerobic digestion, ultrasonication can vary considerably in duration and intensity. For instance, Zawiega found an intensity of 4.3 W $cm^{-2}$ for 300 s to be optimal for solubilizing WAS in treating 0.5 L sludge samples prior to digestion in a 5 L reactor for 28 d [49]. Assuming that the WAS sample was diluted to a volume of 5 L, this represents a power input of 2580 J $L^{-1}$ of WAS. Zhao et al. [47] found optimal waste solubilization for treating WAS with 0.6 W $mL^{-1}$ for 30 min, resulting in a power input of 1080 kJ $L^{-1}$ sludge. Alagöz et al. used 10,000 kJ $kg^{-1}$ VS as a pretreatment to solubilize olive and grape sludge prior to digestion [51].

In the present study, power consumption averaged about 68 W while playing music, and about 140 W when playing a 1000 Hz sine wave. This represents a power consumption of 430 and 886 J $L^{-1}$ $d^{-1}$ digestate, respectively. The loading rate of volatile solids averaged 97 kg $month^{-1}$. At power levels of 68 W and 140 W, this represents a power consumption of approximately 1070 and 2500 kJ $kg^{-1}$ VS $month^{-1}$, respectively. Given the differences in the geometry of ultrasonic probes and audio speakers and the highly variable differences

in digester/reactor volumes between numerous research studies, meaningful comparisons of power consumption versus gains in biogas production are problematic. However, it is likely that the duration of sound played to the digester could be reduced considerably from the 2 h on/1 h off cycle used here to achieve increases in gas production more efficiently.

## 4. Conclusions

Sonic treatment of the anaerobic digestate increased biogas production by 38% compared to the control over the course of the two trial periods, but the only noticeable effect on wastewater malodorants was that sonic treatment reduced the concentration of VFAs. Given that VFAs are precursors to $CH_4$ production, this was not surprising.

Anaerobic digestion has the potential to greatly reduce pollution from wastewater. This is particularly true in the case of animal waste, much of which receives inadequate treatment before its release to the environment. However, anaerobic digestion is slow, and usually employs expensive heating and mixing systems to accelerate biogas production [52,53]. Sonic treatment has the potential to significantly enhance biogas production and potentially replace heating and mixing systems to some degree. However, its influence on odor control appears to be negligible.

**Supplementary Materials:** The following supporting information can be downloaded at: https://www.mdpi.com/article/10.3390/environments9080102/s1, MP3 file S1: 5 min control digester bottom hydrophone; MP3 file S2: 5 min sound-treated digester bottom hydrophone.

**Author Contributions:** Conceptualization, J.L.; methodology, J.L., P.S., N.L. and K.S.; investigation, J.L., N.L., P.S. and K.S.; writing—original draft preparation, J.L.; writing—review and editing: J.L., N.L., P.S. and K.S.; supervision, J.L.; funding acquisition, J.L. and K.S. All authors have read and agreed to the published version of the manuscript.

**Funding:** Funding was provided by the Agricultural Research Service (Grant No. 5040-12630-006-00D).

**Data Availability Statement:** The data presented in this study are available upon request to the corresponding author.

**Acknowledgments:** The authors thank Stacy Antle, Michael Bryant, and Zachary Berry (USDA-ARS) for technical assistance. The use of trade, firm, or corporation names in this website is for the information and convenience of the reader. Such use does not constitute an official endorsement or approval by the United States Department of Agriculture or the Agricultural Research Service of any product or service to the exclusion of others that may be suitable.

**Conflicts of Interest:** The authors declare no conflict of interest. The funders had no role in the design of the study; in the collection, analyses, or interpretation of data; in the writing of the manuscript; or in the decision to publish the results.

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
