# Peer review of "Acoustic Stimulation of Anaerobic Digestion: Effects on Biogas Production and Wastewater Malodors"

_environments, doi:10.3390/environments9080102_

Round 1
Reviewer 1 Report
The paper describes an interesting and innovative measure to improve the performance of anaerobic digesters. The authors focus on two parameters, namely biogas production and removal of odors. These are very relevant but in my opinion neither of the two is adequately elaborated. The authors claim to have proven that sound treatment has a significant effect on biogas production but in this paper they again spend quite some effort on demonstrating the effect of sound on biogas production. However, in my point of view evidence provided in this paper is not conclusive. Moreover, the same data are used as in the previous paper (https://www.mdpi.com/2076-3298/7/2/11) so these data could just have been reported in the Materials and methods section to describe the operation of the digesters. The section on the elimination of odor is longer but contains little data and a lot of speculation. In my opinion the authors should have taken the effort to measure the S containing compounds hydrogen sulfide, methyl mercaptane and dimethyl sulfide, which could not be measured by the selected analysis method but are believed to have a substantial effect on the smell of digestate. Moreover, authors are only measuring in the supernatant and not in the biogas. Can they exclude that volatile compounds escape via the biogas and that effectively the removal of volatile malodors is not affected?
Based on the experimental data provided I find the conclusions premature. However, I have to state that I find it difficult to extract the evidence provide for the effectivity of sound so I may not be the right person to judge this paper. I will also make this clear to the editor.
Please find my detailed comments below
Line 11: why would low frequency sound accelerate biogas production? Elaborate shortly in abstract
Line 12-13: Study period is irrelevant for abstract
Line 35-36: This is the first that I hear that anaerobic systems may be an effective means of reducing malodor. This is a very generalistic statement and the examples given in the text are also very generally described. I can imagine that the effect of AD on odor will largely depend on the compounds under study. The effect of temperature may also be related to higher production of biogas thus lowering the odorous compound concentration, presence of different microorganisms at higher temperatures able to degrade compounds, higher volatility of different compounds.
Line 59 and below: the description of the experimental set up is incomplete. It is unclear whether the speakers are place in the reactor or in the digestate. Furthermore, it is unclear why the authors mention that HRT and SRT are uncoupled. If that is the case, why refer to digestate and not to effluent.
Line 86 and below: I think that an important group of potential compounds was not analysed (sulfides, mercaptans,....). The main reason seems to be that there was no analytical method available, which I think is the wrong reason to ignore the compounds. Moreover, volatile compounds are measured in the supernatant. Why not in the biogas. This could have given a better effect in the insight in the actual effect of sound application on the fate of the selected compounds.
Line 118: section 3.1 This is a replication of data earlier published which should be included in the material and methods section. Given the available data I think it is difficult to draw conclusions about the effect of sound treatment on gas production given the variation in feeding, and HRT. Furthermore, it seems that the gas production in figure 2 more or less follows the VS in the feed (in figure 1). So what is this effect of the 6-12 week HRT anyway. It seems that the easily biodegradable VS is degraded without much delay which could explain the performance of the control reactor, which shows improved performance in time as a response to increased feed. Maybe the microbial population in the reactor adapted to the feed? Can the authors give an approximated of the conversion yield achieved in both reactors? This could give more insight in the performance.
Line 134: The microbial activity is not necessarily higher in the control reactor. The differences in activity may not be that big anyway. There is only a 4% increase in biogas production. One can (at least in my opinion only conclude something about activity if one knows the amount of sludge in the reactor.
Line 163 and below: Contains 3.2-3.4 contain a lot of speculation. If the biogas would have been analysed for the same compounds this could have given more insight in the overall balance between the compounds between gas and liquid phase.
Line 265: section 3.5 Again a lot of speculation. A typo in line 287 “Ver leerdam”?
Line 287: method to track the changes in the concentrations should be described in the Material and Methods section.
Line 304: VFA can also be produced from protein and fats
Line 349: The evidence provided in the supplemental info is unclear to me.
Author Response
The section on the elimination of odor is not really on the elimination of odor but rather a discussion on how odors were not affected by the sound treatment. Some volatile compounds will of course escape through the biogas but as the odors did not differ significantly in the sound and control digesters it could not have happen at significantly different rates. The only claim we make about odor reduction in the paper is that volatile fatty acids were reduced by sound treatment which seems intuitive since biogas production was enhanced by the sound treatment and the volatile fatty acids are precursors to methane production.
We feel that reporting the biogas production in the results section makes sense since we are combining data from two separate research papers covering two separate trials both of which are concurrent with the times the odor samples were analyzed. It helps emphasize that although biogas production was greatly enhanced by sound treatment, odor quality was not notably improved. We feel the speculation is limited and is a discussion of published research directly related to our findings. As discussed in our response below, we did not measure hydrogen sulfide and other very volatile sulfur compounds because of a lack of resources.
Line 11: We added a simple statement to the abstract stating that low frequency sound might biogas production and elaborated on this in lines 353-357 by detailing components such as acoustic streaming , vibrational energy, and cavitational effects.
Line 12-13: We deleted reference to the study periods from the Abstract.
Line 35-36: We elaborated a bit on this to say that odor reductants were determined by Orzi et al using panelists and an olfactometer assay and that in Wilson et al. the reduction of volatile sulfur compounds included methanethiol, dimethyl sulfide, dimethyl disulfide, and dimethyl trisulfide. We also added a reference to Welsh et al. describing odor reduction in pig manure at 25 degrees C. It a valid point that odor compounds might be reduced due to volatilization by the high temperatures as used in Wilson et al. or by greater activity of microorganisms at higher temperatures. Wilson et al. did not address these points specifically and it remains an open question.
Line 59 and below: We made the placement of the speakers underwater clearer, and as per our response to reviewer #2 elaborated on the uncoupling of HRT and SRT. We changed the wording on line 119 to digestate and indicated that the samples were taken from the tank and not the effluent.
Line 86 and below: The main reason we did not analyze the compounds in the vapor phase was not so much as a lack of equipment to do so, but rather a shortage of personnel. We have a small research unit and did not have the time to analyze vapor phase concentrations of the odors in a timely manner while operating the digesters. By nature, vapor phase odor analyses are labile and therefore must be analyzed quickly. By analyzing odors in the supernatant, we were able to store the samples at low temperatures and analyze them later. We agree that it would have been desirable to measure hydrogen sulfide and methyl mercaptan.
Line 118: The variation in biogas production does more or less follow the amount of volatile solids fed, as would be expected. The salient points are that while gas production in both digesters is dependent on the rate of volatile solids fed, the gas production from the sound treated digester is greater than that of the control digester until the volatile solids are depleted in the sound digester. The control digester produces more gas at the end of the experiment simply because it produced so little gas the previous winter. We feel this data is worth showing in the results section as it relates directly to the discussion of results.
There is no effect on the data of the 6-12 week HRT as regards sound treatment of the digesters, nor on VS feeding rate since both the digesters had the same HRT, and VS feeding rate. The VS were obviously degraded much more quickly in the sound treated digester leading to the control digester eventually producing more biogas when the weather warmed in the spring of 2020 whereas the VS had already been consumed in the sound treated digester.
The conversion yields of volatile solids to biogas are given in lines 165-179.
Line 134: Actually, gas production from the sound treated digester was 87% higher than the control digester in 2018 and 22% higher in 2019-2020 even when factoring in the fact that that biogas production dramatically declined in the sound treat digester in June and July 2020 due to volatile solids depletion. Overall, in both trial periods combined, biogas production was 35% higher from the sound treated digester than the control digester.
Line 163 and below: there may be a fair amount of speculation, but we don’t feel that it is overdone. Indeed, given the fact that we didn’t claim that sound treatment effected any change in the concentration of odors in the wastewater we feel that it was hard to speculate too much. It took a fair amount of literature review to compilate all the biosynthetic origins of common wastewater malodors and thought it would be a valuable contribution to the literature to compile all this literature in one paper.
Line 265: We corrected the typo. We speculate some but only in discussing results in light of published research.
Line 287: We tracked changes in the concentrations by stir bar sorptive extraction and discussed this in lines 113-134. We neglected to spell out this term and added it on line 114. We also elaborated quite a bit on how odor compounds and the greenhouse gases were quantified. We apologize for not doing this adequately in the original submission.
Line 304: True, but quantitatively VFA are mainly the product of carbohydrate fermentation. That is why we used the word “largely” in the first sentence of that paragraph. In addition to fatty acids and proteins, VFA can also be formed in great amounts (acetic acid) by hydrogenotrophic acetogens.
Line 349: We modified this paragraph (also see response to reviewer #2) to say the louder background audio levels in the sound treated digester was evidence of greater gas production in that bubbles act as acoustic resonators and produce sound.
We thanks the reviewer for their thoughful comments.
Reviewer 2 Report
This study investigated acoustic stimulation of AD, focusing on biogas yield and malodors. The experiment was well conducted with a range of odor generating compounds quantified and discussed. The research is presented clearly, although a few aspects should be further clarified, as below.
1. it is unclear why the malodors refer to wastewater malodors, considering agricultural waste and residues were used in the study rather than wastewater.
2. Lines 66-73 should be better organised to explain how the hydraulic retention time was decoupled from solids retention time.
3. The methods to quantify and report biogas production and composition (lines 128-135, and the paragraphs below that) should be clarified, including under what temperature and pressure condition the biogas volume is reported. Based on the concentrations of CO2 and CH4 for control and sound treated digester, the biogas pressure of the two digesters were different. The volumes under different pressure cannot be compared directly for the purpose of biogas production comparison.
4. Lines 348-350: it is not clear why sludge disruption facilitates microbial colonisation of the sludge, also it is not clear why this is supported by the supplemental material. Further clarification is needed. This should be further explained.
Author Response
As the reviewer noted, the material and methods section could be improved and we added more details on CO2, CH4, and odor compound sampling, identification, and quantitation. Per the reviewers’ comments we edited the results to make them (hopefully) easier to follow.
- We modified the first sentence of the abstract to reflect that wastewater malodors refers to the malodors generated by digestion of agricultural waste and residuals.
- We changed the last sentence of the first paragraph of section 2.1 to read “The digesters therefore had an HRT of approximately 12 weeks when fed once weekly and about 6 weeks when fed twice weekly. The wastewater outlet from the digesters was approximately one meter above the bottom and therefore given settling of solids, designed to uncouple sludge retention time from wastewater retention time.” to better explain how hydraulic retention time was decoupled from solids retention time.
- We added brief descriptions of how the gas production and gas quality were measured (lines 113-188), and we modified paragraph 2 of section 3.1 was modified to show that the pressures were likely different in the two digesters.
- We modified this paragraph (L349-255) to say the higher background sound levels in the sound treated were likely due to higher gas production and how sound treatment can enhance microbial colonization. We also added a statement that bubbles act as acoustic resonators counting as evidence for greater gas production being evidence for greater microbial colonization.
Thank you for your help.
Reviewer 3 Report
General:
The introduction section must be improved by a literature review regarding the paper’s subject. I suggest that some parts of the following chapters (3-3 to 3-6) are moved to “Introduction”. Chapter 3.1 must be improved and rewritten, while the next chapters are much better. There is a gap between the first part of the paper in chapter 3-1 and other chapters. Some connection is required to understand how biogas production is in relationship to malodors with and without acoustic stimulation.
Errors:
L37: Insert literature number for »Orzi et al.«
L131-132: something is wrong with units mmol and m-3, please check.
L157-161: explain more in detail, what does this increase of VS in control compared to the sample mean? From this paragraph, it could be concluded that the biogas yield is decreasing with the sound, but authors claimed that yield increased? Please explain more clearly.
Author Response
We did slightly expand the literature review in the Introduction to expand upon the idea that since elevated temperatures enhance biogas production and can improve malodors maybe sound treatment might also enhance malodor reductions since it also enhances biogas production. We would prefer to leave the literature review in the results and discussion as it is since we like the way the individual sections discuss biosynthetic origins and whether or not the sound treatment affects the concentrations of the malodors. We also rewrote parts of chapter 3.1 to make the explanation of volatile solids depletion in the sound treated digester clearer.
L27, we moved [3] to after Orzi et al. instead of the end of the sentence.
L131-132. The units are correct, but we made a typo in placement. we changed the line to read “Average weekly gas production over this period was 87% higher from the sound treated digester (11.9 m3 wk-1) than the control digester (6.4 m3 wk-1) and the concentration of CO2 and CH4 averaged 10,700 l and 36,800 mmol m-3 for the control digester and 11,100 and 38,200 mmol m-3 for the sound treated digester from early July to mid-October [4], since we misplaced the words mmol m-3 after 10,700 instead of 36,800.
Line157-161: We rewrote this paragraph to make it clearer that by decoupling solids retention from hydraulic retention time that the retained volatile solids were depleted in the sound treated digester more so than in the control digester and therefore there was less gas produced in the sound treated in the last two months.
Thank you for your help.